# Effect of Household Type on the Prevalence of Climacteric Syndrome among Middle-Aged Men

**DOI:** 10.3390/healthcare11192684

**Published:** 2023-10-05

**Authors:** Dohhee Kim, Seunghee Lee, Mijung Jang, KyooSang Kim

**Affiliations:** 1Department of Research Institute, Seoul Medical Center, Seoul 02053, Republic of Korea; dohheekim@seoulmc.or.kr (D.K.); shlee282@seoulmc.or.kr (S.L.); mijungjang@seoulmc.or.kr (M.J.); 2Department of Occupational Environmental Medicine, Seoul Medical Center, Seoul 02053, Republic of Korea

**Keywords:** household type, climacteric syndrome, middle aged, dietary factors

## Abstract

Research on climacteric syndrome among middle-aged men remains scant compared to the research among women. Research is also lacking on climacteric syndrome among older adults living alone, particularly men, who are more vulnerable than females living alone. This cross-sectional study investigated whether the prevalence of climacteric syndrome is associated with the type of household middle-aged men live in and identified the determinants of climacteric syndrome based on the household type. Six hundred middle-aged men living in multi-person households and six hundred living alone were surveyed about general characteristics, diet-related factors, and climacteric syndrome. Data were analyzed using Pearson’s chi-squared test, Fisher’s exact test, and logistic regression. The risk of climacteric syndrome in single-person households was found to be 1.6 times higher than that among multi-person households (*p* = 0.006). In multi-person households, income and breakfast frequency predicted climacteric syndrome (*p* < 0.05), while age, breakfast frequency, dinner frequency, and weekly eating out frequency predicted climacteric syndrome in single-person households (*p* < 0.05). Thus, dietary factors are more closely linked to the prevalence of climacteric syndrome in single-person households than in multi-person households. This highlights the need for climacteric syndrome interventions for middle-aged men, whose health concerns may persist into older adulthood.

## 1. Introduction

Single-person households comprise only one individual. This concept differs from the concept of “single” individuals, and such households are increasing globally [1]. In the 2010s, the average number of single-person households in Asia was lower than those in Europe and North America. However, since then, the number of single-person households has grown most rapidly in Korea among the OECD countries [2]. The reasons for this increase vary widely. They include later first marriages, decreased marriage rates, increased divorce rates, an increased number of single-person households, and an increased number of older adults living alone due to population aging [3]. Single-person households hold significance in the social structure not only in terms of their socioeconomic aspects but also their health aspects.

In Republic of Korea, 44% of single-person households comprise middle-aged adults (individuals aged between 40 and 64), and social isolation, a concern considered unique to older adults living alone, also affects middle-aged adults [4]. Middle adulthood is a crucial phase that exposes individuals to various negative experiences, such as stress, climacteric syndrome, and depression. It also makes them vulnerable to social problems such as alcohol abuse, marital discord, divorce, and social isolation [5]. Notably, an increasing number of middle-aged men live alone and face socioeconomic difficulties. The challenges individuals face in middle adulthood also affect them in later years, which further emphasizes the importance of middle adulthood [6].

Male climacteric syndrome is also known as late-onset hypogonadism and testosterone deficiency syndrome. It is reported that, in Korea, the prevalence of climacteric syndrome among middle-aged men exceeds 60% [7,8]. While female climacteric syndrome is characterized by unique hormonal changes, male climacteric syndrome does not present clear physical changes. Resultingly, research on female climacteric syndrome is abundant, while research on male climacteric syndrome remains scarce [9,10].

The household type and climacteric syndrome are two crucial determinants of the mental and physical health of middle-aged men. These men are often socially and economically vulnerable, and effectively managing climacteric syndrome may benefit their health in later years. Thus, this study investigates whether the prevalence of climacteric syndrome is associated with the type of household middle-aged men live in and identifies the factors of climacteric syndrome among middle-aged men based on the household type. We assess climacteric syndrome based on aging males’ symptoms, such that climacteric syndrome is diagnosed if symptoms suggest hormonal deficiency. Consequently, we use the aging males’ symptoms scale to determine the existence of climacteric syndrome [11]. This study identifies actionable factors associated with the male climacteric in middle adulthood. It shows that household type and dietary factors are crucial factors in reducing the prevalence of climacteric syndrome among middle-aged men.

## 2. Materials and Methods

We surveyed middle-aged men (men aged between 40 and 64) residing in Seoul, Korea, between October 2022 and November 2022. This survey was administered by a professional survey company using self-report online and offline questionnaires. All middle-aged men registered with a professional survey company were asked to participate in the study. To evenly distribute the age of the participants, we conducted an additional offline survey of participants who were not registered with a professional survey company. Only those individuals who provided informed consent were enrolled. The questionnaire was evenly distributed among men aged between 40 and 44, 45, and 49, 50 and 54, 55, and 59, and 60 and 64. A total of 1832 individuals completed the survey. Among these, 46 individuals dropped out, 209 failed the criterion regarding age and household type, and 284 provided the same answers to questions. Additionally, to equalize the two groups, we sequentially excluded the responses of 93 individuals whose response time was less than the average response time. Resultingly, the responses of 1200 participants—600 living in multi-person households and 600 living in single-person households—were included in the data analysis. Furthermore, the study was approved by the Institutional Review Board of Seoul Medical Center (IRB 2022-07-006).

After obtaining permission from its developer, we used the Korean version of the aging males’ symptoms (AMS) scale to measure male climacteric syndrome and quality of life [12]. The AMS scale is a health-related quality of life scale (HRQoL) for men that was developed to self-administrate their symptoms. The scale comprises seventeen items, with seven items measuring somatic symptoms, five measuring psychological symptoms, and five measuring sexual symptoms. The severity of each symptom is rated on a 1–5 scale. AMS is divided into four severity categories: no or few symptoms (17–26), mild symptoms (27–36), moderate symptoms (37–49), and severe symptoms (50 or more). In the AMS subscale, the severity of the psychological subscale and sexual subscale is the same: no symptoms (0–5), mild symptoms, or more (6 or more). The somatic subscale has no symptoms (0–8) and mild symptoms (9 or more). In this study, climacteric syndrome was diagnosed if the total score was 27 (mild symptoms) or higher [13]. In addition, the AMS subscale for climacteric syndrome was diagnosed as mild symptoms or more, the same as the AMS scale diagnosis [11].

We also collected data on participants’ characteristics, including age, education level, occupation, income level, whether they smoke, drinking behavior, physical activity, sleep duration, and body mass index (BMI). The income level was classified as high- and low-income levels based on a median of KRW 50 million. Professional, management, office, and service work were considered white-collar occupations, while technical work, agricultural work, manual labor work, and being unemployed were considered blue-collar occupations. Drinking alcohol more than two times a week and having, on average, seven drinks or more in each sitting was considered high-risk drinking behavior. Not indulging in such drinking behavior was considered low-risk drinking behavior. Indulging in moderate-intensity aerobic physical activity for at least 150 min or vigorous-intensity aerobic physical activity for at least 75 min was considered adequate physical activity. Not having this much physical activity was considered inadequate physical activity. A sleep duration of more than six hours was considered sufficient sleep duration. Furthermore, one whose BMI equaled or exceeded 25 was considered obese. We categorized these variables based on the guidelines prescribed by the World Health Organization and National Sleep Foundation [14,15]. Dietary factors were surveyed to examine the influence of lifestyle-related factors. They included the frequency of breakfast, lunch, and dinner, having company when eating, and the frequency of eating out each week.

Statistical analyses were performed using SPSS version 20 (IBM Corp., Armonk, NY, USA). We compared lifestyle factors based on household type and climacteric syndrome using Pearson’s chi-squared test and Fisher’s exact test. The effects of general characteristics and dietary factors on climacteric syndrome were analyzed using logistic regression.

## 3. Results

### 3.1. General Characteristics Associated with Climacteric Syndrome

The prevalence of climacteric syndrome differed significantly based on participants’ age, education level, occupation, income level, smoking status, and household type. In terms of socioeconomic factors, the prevalence of climacteric syndrome was significantly higher among participants with lower education levels, manual labor work, and lower income. In terms of health-related factors, the prevalence of climacteric syndrome differed significantly based only on participants’ smoking status. In terms of household type, the prevalence of climacteric syndrome was significantly higher among the participants living in single-person households than those living in multi-person households (Table 1).

The AMS scale consists of three subscales: psychological, somatic, and sexual. To further investigate the association of general characteristics with climacteric syndrome using the AMS scale, the results of the analysis of general characteristics according to the AMS subscale are shown in Appendix A. The psychological subscale was significant with education level and household type. The somatic subscale was significantly related to age, education level, smoking status, and household type. The sexual subscale was significant only for age and physical activity.

### 3.2. Risk of Climacteric Syndrome Based on General Characteristics

We performed bivariate logistic regression with the general characteristics that were significantly related to climacteric syndrome in the univariate analysis as the independent variables and climacteric syndrome as the dependent variable. Climacteric syndrome differed significantly based on participants’ age, education level, and household type. More specifically, the risk of climacteric syndrome was nearly 1.4 times higher among men aged 50 or more than those aged less than 50 (OR = 1.376, 95% CI 1.035–1.829, *p* = 0.028) and approximately 1.6 times higher among high school graduates than among college graduates (OR = 1.557, 95% CI 1.006–2.409, *p* = 0.047). In terms of household type, the risk of climacteric syndrome was nearly 1.5 times higher among the participants living in single-person households than those living in multi-person households (OR = 1.515, 95% CI 1.125–2.038, *p* = 0.006) (Table 2).

### 3.3. General Characteristics Associated with Climacteric Syndrome Based on Household Type

Since the household type and the prevalence of climacteric syndrome were found to be significantly associated, we examined the general characteristics related to the prevalence of climacteric syndrome separately for single-person and multi-person households. In multi-person households, the prevalence was significantly higher among low-income men and smokers. In single-person households, the prevalence was significantly higher among older and less-educated men (Table 3).

### 3.4. Dietary Factors Associated with Climacteric Syndrome Based on Household Type

Like the results obtained from examining participants’ general characteristics, the prevalence of climacteric syndrome differed significantly based on the household type when we examined participants’ dietary factors. In multi-person households, the prevalence was significantly higher among those who eat breakfast four or fewer times a week than those who eat breakfast five or more times a week. No other significant differences were found in the prevalence of climacteric syndrome based on dietary factors (*p* < 0.05). However, in single-person households, the prevalence of climacteric syndrome was significantly higher among those who eat breakfast and dinner four or fewer times a week and those who eat out three or more times a week (Table 4).

The results of dietary factor analysis using the AMS subscale are shown in Appendix A. In the psychological and somatic subscales, the same significant differences were found in the breakfast frequency, shared breakfast, dinner frequency, shared dinner, and eating. In contrast, the sexual subscale was significant only in the dinner frequency.

### 3.5. Factors of Climacteric Syndrome Based on Household Type

Table 5 shows the results of bivariate logistic regression, which determined the factors of climacteric syndrome based on household type using the general characteristics and dietary factors associated with the prevalence of climacteric syndrome in univariate analyses. In multi-person households, the risk of climacteric syndrome was 1.5 times higher among low-income men than among high-income men (adj. OR = 1.553, 95% CI 1.031–2.339, *p* = 0.035) and 1.6 times higher among those who eat breakfast four or fewer times a week than those who eat breakfast five or more times a week (adj. OR = 1.659, 95% CI 1.131–2.433, *p* = 0.010). In single-person households, the risk of climacteric syndrome was nearly 1.9 times higher among men aged 50 or more compared to those aged less than 50 (adj. OR = 1.871, 95% CI 1.199–2.919, *p* = 0.006). Furthermore, the risk of climacteric syndrome was 1.8 times higher among those who eat breakfast four or fewer times a week than those who eat breakfast five or more times a week (adj. OR = 1.806, 95% CI 1.099–2.970, *p* = 0.020). The risk was also 2.2 times higher among those who eat dinner four or fewer times a week than those who eat dinner five or more times a week (adj. OR = 2.240, 95% CI 1.293–3.882, *p* = 0.004). In terms of the frequency of eating out, the risk of climacteric syndrome among those who eat out at least three times a week was nearly double the risk among those who eat out two or fewer times a week (adj. OR = 1.991, 95% CI 1.276–3.106, *p* = 0.002) (Table 5).

## 4. Discussion

Unlike in the past, household structures have diversified, making socioeconomic, physical, and mental health issues concerning single-person households salient concerns. The characteristics of single-person households vary across generations. Young adults tend to live alone voluntarily because of work and school. However, middle-aged adults are likely to have come to live alone due to reasons such as divorce and widowhood and will probably continue to live alone in later years. Furthermore, single-person households tend to have a lower household income and education level than multi-person households, and education level, along with age, is closely associated with the prevalence of climacteric syndrome [16]. In this study, age, education level, and household type were found to be associated with the prevalence of climacteric syndrome among middle-aged men in Korea, suggesting that the household type may be as important as age and education level in the development of climacteric syndrome. The association of socioeconomic factors with climacteric syndrome was further confirmed by analyzing the AMS subscales of psychological, somatic, and sexual aspects. The somatic subscale showed the strongest associations with socioeconomic and health-related factors, and each subscale showed different patterns of association with each factor. These results suggest that each AMS subscale diagnoses each distinct aspect of climacteric syndrome.

Climacteric syndrome refers to the physical, mental, and sexual changes that naturally occur during aging. The male climacteric is diagnosed based on declining physical abilities, such as energy, muscle strength, endurance, and sexual decline, and mental health problems, such as depression, anxiety, and increased stress [17]. Unlike climacteric syndrome in women, the male climacteric is not accompanied by pronounced physical or physiological changes. Thus, it is difficult to recognize and should be diagnosed based on a multidimensional assessment examining physical, mental, and physiological aspects. The AMS scale used in this study evaluates not only sexual factors, but also physical and psychological factors related to changes in the male climacteric, which increases its usefulness for diagnosing climacteric syndrome [18]. The AMS scale can assess aging symptoms in men regardless of disease, assess symptom severity over time, and measure changes before and after androgen treatment. As such, it is one of the leading standardized measures of the severity of climacteric syndrome in men [13].

In this study, the results of dietary factors using the AMS scale showed that breakfast, dinner, and eating out may be associated with climacteric syndrome. In addition, the analysis using the AMS subscale showed that dietary factors were highly associated with the psychological and somatic aspects of climacteric syndrome. While lifestyle differences between single-person and multi-person households cannot be solely diet-based, diet is an important feature distinguishing different types of households. It is reported that, compared to high-income individuals, low-income individuals have a poorer standard of living, are at an elevated risk of stress and depression, and are likely to fall into a negative life cycle by spending a large percentage of their income on meeting basic needs, such as food [19]. Studies have also found that single-person households spend a large percentage of their income on buying food and tend to purchase more processed foods because of their convenience and simple use [20]. Notably, this study found that differences in the dietary factors of households are also linked to the prevalence of climacteric syndrome. It also discovered that the risk of developing climacteric syndrome nearly doubles when the frequency of eating out is at least three times a week compared to when one eats out less than three times weekly in single-person households. A previous study reported that single-person households tend to skip breakfast more frequently than multi-person households and *honbap* (eating alone) leads to excessive sodium intake and nutritional imbalance, which affects health [21]. However, we found that skipping dinner is a more critical risk factor for climacteric syndrome than skipping breakfast. This may be because the syndrome is diagnosed after assessing not only one’s physical health but also one’s mental and sexual health [18]. If left unaddressed, sustaining such dietary habits may harm the health of middle-aged adults, which may hamper their health in older adulthood. In fact, the 2014 National Survey of Older Koreans showed that 24% of older adults living alone skip breakfast, which is higher than the percentage among older adults living with a spouse (10%) and older adults living with their children (11.2%) [22].

In addition to their nutritional aspect, dietary factors are closely linked to mental health indicators, such as stress, sleep, and depression. Eating alone has already been established as a factor of depression, and dining with family members is known to facilitate family bonding and social support and maintain mental health [23]. Furthermore, eating dinner alone significantly increases the rate of depressive mood and suicidal ideation, with the effects being more prevalent among men [24]. However, unlike the results of previous studies, we did not find a statistically significant relationship between sharing a meal and the prevalence of climacteric syndrome. This may be because the mental health aspect of the AMS scale is not appropriate or inadequate for assessing overall mental health.

This study was conducted by a professional survey company and consisted mostly of participants who were registered with the survey company. Although a small number of unregistered participants were also included in the study to ensure an even distribution of age, factors such as education, income, and occupation could not be controlled for and may not be representative due to sampling bias. As climacteric syndrome is highly associated with socioeconomic factors, there are limitations to drawing general conclusions from the study. Another limitation of this study is that we could not examine the effects of nutrient intake and the loneliness and depression caused by eating alone. Therefore, it is necessary to further study how different dietary factors in different household types affect nutritional deficiencies and health and to consider including them in national surveys to ensure representative data.

## 5. Conclusions

Household type and dietary factors cannot be conclusively established as risk factors for climacteric syndrome owing to the variations in diet based on customs and race. Nevertheless, the type of household and the consequent differences in dietary factors may be crucial factors for middle-aged men, not only in preventing climacteric syndrome but also in managing health. Barring non-modifiable or less-modifiable risk factors, such as age, education level, and income, household type and dietary factors may be considered vital when reducing the prevalence of climacteric syndrome among middle-aged men. Therefore, this study identified potential intervenable factors associated with the male climacteric in middle adulthood, a period in which individuals have financial independence and can engage in activities that improve health. This study is also meaningful because it presents actionable risk factors that may also have an impact on the mental and physical health of older adults, particularly those living alone.

## Figures and Tables

**Table 1 healthcare-11-02684-t001:** General characteristics associated with climacteric syndrome.

Variable		Normal	Climacteric Syndrome	*p*
		*n* (%)	*n* (%)	
Age	Less than 50	114 (26.6)	315 (73.4)	0.016 *
	50 or more	158 (20.5)	613 (79.5)	
Education level	University	241 (24.7)	736 (75.3)	0.001 **
	High school	31 (13.9)	192 (86.1)	
Occupation	White-collar	228 (24.0)	723 (76.0)	0.034 *
	Blue-collar	44 (17.7)	205 (82.3)	
Income level	High	156 (25.9)	446 (74.1)	0.007 **
	Low	116 (19.4)	482 (80.6)	
Smoking status	Non-smoker	172 (24.9)	520 (75.1)	0.035 *
	Smoker	100 (19.7)	408 (80.3)	
Drinking behavior	Low-risk	214 (22.5)	738 (77.5)	0.761
	High-risk	58 (23.4)	190 (76.6)	
Physical activity	Adequate	58 (26.0)	165 (74.0)	0.186
	Inadequate	214 (21.9)	763 (78.1)	
Sleep duration	Sufficient	135 (22.5)	466 (77.5)	0.866
	Insufficient	137 (22.9)	462 (77.1)	
BMI	Normal	159 (23.1)	528 (76.9)	0.648
	Obesity	113 (22.0)	400 (78.0)	
Household type	Multi-person	166 (27.7)	434 (72.3)	<0.001 ***
	Single-person	106 (17.7)	494 (82.3)	

* *p* < 0.05, ** *p* < 0.01, *** *p* < 0.001.

**Table 2 healthcare-11-02684-t002:** Factors associated with climacteric syndrome.

Variable		B	SE	Wald	*p*	OR	95% CI	
							LLCI	ULCI
Age	50 or more	0.319	0.145	4.833	0.028 *	1.376	1.035	1.829
Education level	High school	0.443	0.223	3.953	0.047 *	1.557	1.006	2.409
Occupation	Blue-collar	0.058	0.199	0.085	0.770	1.060	0.718	1.564
Income level	Low	0.190	0.150	1.596	0.206	1.209	0.901	1.622
Smoking status	Smoker	0.206	0.146	1.998	0.158	1.229	0.923	1.634
Household type	Single-person	0.415	0.152	7.505	0.006 **	1.515	1.125	2.038

SE: standard error, OR: odd ratio, CI: confidence interval, LLCI: lower-limit confidence interval, ULCI: upper-limit confidence interval. * *p* < 0.05, ** *p* < 0.01.

**Table 3 healthcare-11-02684-t003:** General characteristics associated with climacteric syndrome based on household type.

Variable		Multi-Person Household			Single-Person Household		
		Normal	Climacteric Syndrome	*p*	Normal	Climacteric Syndrome	*p*
		*n* (%)	*n* (%)		*n* (%)	*n* (%)	
Age	Less than 50	60 (28.0)	154 (72.0)	0.880	54 (25.1)	161 (74.9)	<0.001 ***
	50 or more	106 (27.5)	280 (72.5)		52 (13.5)	333 (86.5)	
Education level	University	155 (28.3)	392 (71.7)	0.239	86 (20.0)	344 (80.0)	0.017 *
	High school	11 (20.8)	42 (79.2)		20 (11.8)	150 (88.2)	
Occupation	White-collar	147 (28.4)	371 (71.6)	0.327	81 (18.7)	352 (81.3)	0.282
	Blue-collar	19 (23.2)	63 (76.8)		25 (15.0)	142 (85.0)	
Income level	High	119 (30.5)	271 (69.5)	0.034 *	37 (17.5)	175 (82.5)	0.919
	Low	47 (22.4)	163 (77.6)		69 (17.8)	319 (82.2)	
Smoking status	Non-smoker	119 (30.4)	273 (69.6)	0.043 *	53 (17.7)	247 (82.3)	1.000
	Smoker	47 (22.6)	161 (77.4)		53 (17.7)	247 (82.3)	
Drinking behavior	Low-risk	136 (28.0)	349 (72.0)	0.674	78 (16.7)	389 (83.3)	0.246
	High-risk	30 (26.1)	85 (73.9)		28 (21.1)	105 (78.9)	
Physical activity	Adequate	36 (29.8)	85 (70.2)	0.566	22 (21.6)	80 (78.4)	0.257
	Inadequate	130 (27.1)	349 (72.9)		84 (16.9)	414 (83.1)	
Sleep duration	Sufficient	82 (28.7)	204 (71.3)	0.600	53 (16.8)	262 (83.2)	0.570
	Insufficient	84 (26.8)	230 (73.2)		53 (18.6)	232 (81.4)	
BMI	Normal	100 (28.7)	248 (71.3)	0.492	59 (17.4)	280 (82.6)	0.848
	Obesity	66 (26.2)	186 (73.8)		47 (18.0)	214 (82.0)	

* *p* < 0.05, *** *p* < 0.001.

**Table 4 healthcare-11-02684-t004:** Dietary factors associated with climacteric syndrome based on household type.

Variable		Multi-Person Household			Single-Person Household		
		Normal	Climacteric Syndrome	*p*	Normal	Climacteric Syndrome	*p*
		*n* (%)	*n* (%)		*n* (%)	*n* (%)	
Breakfast frequency	Five or more times a week	92 (33.1)	186 (66.9)	0.006 **	33 (28.2)	84 (71.8)	0.001 **
	Four or fewer times a week	74 (23.0)	248 (77.0)		73 (15.1)	410 (84.9)	
Shared breakfast	Together	86 (30.6)	195 (69.4)	0.131	8 (16.0)	42 (84.0)	0.747
	Alone	80 (25.1)	239 (74.9)		98 (17.8)	452 (82.2)	
Lunch frequency	Five or more times a week	146 (28.9)	360 (71.1)	0.132	88 (17.0)	430 (83.0)	0.274
	Four or fewer times a week	20 (21.3)	74 (78.7)		18 (22.0)	64 (78.0)	
Shared lunch	Together	123 (29.6)	292 (70.4)	0.106	70 (16.4)	357 (83.6)	0.199
	Alone	43 (23.2)	142 (76.8)		36 (20.8)	137 (79.2)	
Dinner frequency	Five or more times a week	150 (28.6)	374 (71.4)	0.168	87 (21.2)	324 (78.8)	0.001 **
	Four or fewer times a week	16 (21.1)	60 (78.9)		19 (10.1)	170 (89.9)	
Shared dinner	Together	142 (28.9)	350 (71.1)	0.162	37 (16.3)	190 (83.7)	0.493
	Alone	24 (22.2)	84 (77.8)		69 (18.5)	304 (81.5)	
Eating out	Two or fewer times a week	107 (28.3)	271 (71.7)	0.647	57 (23.9)	181 (76.1)	0.001 **
	Three or more times a week	59 (26.6)	163 (73.4)		49 (13.5)	313 (86.5)	

** *p* < 0.01.

**Table 5 healthcare-11-02684-t005:** Factors of climacteric syndrome based on household type.

Variable		Multi-Person Household				Single-Person Household			
		OR	95% CI		*p*	OR	95% CI		*p*
			LLCI	ULCI			LLCI	ULCI	
Age	50 or more	1.155	0.775	1.721	0.480	1.871	1.199	2.919	0.006 **
Education level	High school	1.131	0.547	2.338	0.739	1.520	0.857	2.696	0.152
Occupation	Blue-collar	1.293	0.719	2.328	0.391	1.096	0.630	1.905	0.746
Income level	Low	1.553	1.031	2.339	0.035 *	1.049	0.654	1.682	0.844
Smoking status	Smoker	1.461	0.979	2.182	0.063	0.889	0.573	1.379	0.599
Breakfast frequency	Four or fewer times a week	1.659	1.131	2.433	0.010 *	1.806	1.099	2.970	0.020 *
Dinner frequency	Four or fewer times a week	1.370	0.752	2.496	0.303	2.240	1.293	3.882	0.004 **
Eating out	Three or more times a week	1.204	0.818	1.772	0.346	1.991	1.276	3.106	0.002 **

* *p* < 0.05, ** *p* < 0.01.

## Data Availability

The data that support the findings of this study are not openly available due to reasons of sensitivity and are available from the corresponding author upon reasonable request.

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
