# Peer review of "Effect of Household Type on the Prevalence of Climacteric Syndrome among Middle-Aged Men"

_healthcare, 2023, doi:10.3390/healthcare11192684_

Round 1
Reviewer 1 Report
Review Report for: “Effect of the household type on the prevalence of climacteric syndrome among middle-aged men.”
Summary
This cross-sectional study examined household characteristics and meal patterns on the rates of climacteric syndrome in Korean men. There are limited studies on this topic and even fewer on household types.
General concept comments
1. Overall, the article is well written with a clearly defined purpose. The authors address inconsistencies in the existing literature and provide insights into the benefits of their study results.
Specific Comments to be Addressed
1. In the Materials and Methods section – add how the participants were selected to take the questionnaire.
2. Line 80 – spell out what the acronym AMS stands for.
Reviewer 2 Report
There are two limitations that need to be discussed in the Discussion Section.
1-1. Since this study is cross-sectional in design, causality should be cautiously explained.
1-2. The data used in this study were obtained from an Internet research company. The sample may not be representative of the entire Korean population, as it consists of individuals who are part of an online panel. This could introduce sampling bias, potentially excluding individuals who are less likely to participate in online surveys, such as those with limited internet access (e.g., low income, as discussed in the manuscript) or mental conditions due to their climacteric syndrome.
Personally, I recommend using the term "factors" rather than "predictors" when describing the statistically significant variables, given the cross-sectional design.
Reviewer 3 Report
In this study, the authors explore the intriguing relationship between household type and the prevalence of climacteric syndrome among middle-aged men. The research sheds light on the potential influence of the domestic environment on this particular syndrome, providing valuable insights into the broader understanding of men's health during middle age.
The study successfully identifies a correlation between household type and the occurrence of climacteric syndrome, highlighting the need for further investigation into the underlying mechanisms and contributing factors. The authors have conducted a thorough analysis, presenting statistically significant findings that add to the existing body of knowledge in this field.
However, to enhance the comprehensiveness of this study, it is essential for the authors to elaborate on the limitations encountered during the research process. Addressing these limitations could refine the study's methodology and further strengthen the validity and applicability of the results. A more in-depth discussion regarding potential confounding variables and the generalizability of the findings to diverse populations would enrich the academic contribution of this research.
Reviewer 4 Report
Introduction:
While studies targeting middle-aged women are frequently encountered in the literature, research focusing on middle-aged men remains remarkably scarce. This scarcity underscores the significance of the present study.
Materials and Methods:
The study appears to have employed a rational sampling method, and there's evidence of ethical considerations. However, the description pertaining to the Aging Males' Symptoms (AMS) scale, which is arguably the most crucial metric in this research, is notably lacking. A more detailed explanation of this scale is recommended. Additionally, there is an absence of specific survey results regarding the AMS scale. Including these results in the findings will significantly aid in understanding the research better.
Results:
All the outcome variables present intrigue. Of particular interest are the significant findings related to dietary factors associated with andropause symptoms based on household type, especially the items focusing on breakfast, dinner, and dining out.
Discussion:
The discussion section could benefit from further elaboration to aptly explain the captivating results. Notably, the absence of specific results related to the AMS scale limits a comprehensive understanding of the study.
